# Swiss-wide multicentre evaluation and prediction of core outcomes in arthroscopic rotator cuff repair: protocol for the ARCR_Pred cohort study

Laurent Audigé [1,2,3] Heiner C C Bucher,[3] Soheila Aghlmandi,[3] Thomas Stojanov,[1,2,3] David Schwappach [4,5] Sabina Hunziker,[6] Christian Candrian,[7] Gregory Cunningham,[8,9] Holger Durchholz,[10] Karim Eid,[11] Matthias Flury,[12] Bernhard Jost,[13] Alexandre Lädermann,[9,14] Beat Kaspar Moor,[15] Philipp Moroder,[16] Claudio Rosso,[17] Michael Schär,[18] Markus Scheibel,[16,19] Christophe Spormann,[20] Thomas Suter,[21] Karl Wieser,[22] Matthias Zumstein,[23,24] ARCR_Pred Study Group, Andreas M Müller[2]

For numbered affiliations see end of article.

**Correspondence to**
Professor Laurent Audigé;
laurent.audige@unibas.ch

## ABSTRACT

**Introduction** In the field of arthroscopic rotator cuff repair (ARCR), reporting standards of published studies differ dramatically, notably concerning adverse events (AEs). In addition, prognostic studies are overall methodologically poor, based on small data sets and explore only limited numbers of influencing factors. We aim to develop prognostic models for individual ARCR patients, primarily for the patient-reported assessment of shoulder function (Oxford Shoulder Score (OSS)) and the occurrence of shoulder stiffness 6 months after surgery. We also aim to evaluate the use of a consensus core event set (CES) for AEs and validate a severity classification for these events, considering the patient's perspective.

**Methods and analysis** A cohort of 970 primary ARCR patients will be prospectively documented from several Swiss and German orthopaedic clinics up to 24 months postoperatively. Patient clinical examinations at 6 and 12 months will include shoulder range of motion and strength (Constant Score). Tendon repair integrity status will be assessed by ultrasound at 12 months. Patient-reported questionnaires at 6, 12 and 24 months will determine functional scores (subjective shoulder value, OSS), anxiety and depression scores, working status, sports activities, and quality of life (European Quality of Life 5 Dimensions 5 Level questionnaire). AEs will be documented according to a CES. Prognostic models will be developed using an internationally supported regression methodology. Multiple prognostic factors, including patient baseline demographics, psychological, socioeconomic and clinical factors, rotator cuff integrity, concomitant local findings, and (post)operative management factors, will be investigated.

**Ethics and dissemination** This project contributes to the development of personalised risk predictions for supporting the surgical decision process in ARCR. The consensus CES may become an international reference for the reporting of complications in clinical studies and registries. Ethical approval was obtained on 1 April 2020 from the lead ethics committee (EKNZ, Basel, Switzerland;

---

**Strengths and limitations of this study**

► This is a large prospective multicentre observation of routine care.
► The study is an assessment of patient-reported outcomes.
► The study will implement an international core outcome set of adverse events.
► The study will use internationally supported methodology for prognostic model development.
► There is potential limited response to patient questionnaires at 24 months.

---

ID: 2019-02076). All participants will provide informed written consent before enrolment in the study.
**Trial registration number** NCT04321005.
**Protocol version** Version 2 (13 December 2019).

## INTRODUCTION

Rotator cuff tears are one of the most common injuries of the shoulder joint, which may cause pain and disability associated with severe restrictions in daily activities. Surgical repair is indicated when non-operative treatment fails or follows extended traumatic tears, notably inactive patients without signs of advanced tendon degeneration or muscle fat infiltration.[1] Clinical studies have demonstrated clinically relevant improvement in shoulder function and quality of life after arthroscopic rotator cuff repair (ARCR).[2–5] The number of ARCRs has increased over the last two decades[6–9] due to several contributing factors, such as an ageing yet active population, improvements

in operative repair techniques and more liberal indications for ARCRs.

Not all patients, however, benefit from ARCR.[10] Patients may be affected by complications and/or adverse events (AEs) such as persistent pain, shoulder stiffness, infection, neurological problems and repair failures.[11 12] About 20% of patients may show, typically between 6 and 12 months following ARCR, a persistent rotator cuff defect.[13] Patients with healed tendons may show better functional outcome after repair.[2 14 15] Postoperative shoulder stiffness, a major complication reported to occur in 1.5%–11.1% of ARCRs,[11] leads to limitations in everyday activities, prolonged rehabilitation, and in severe cases to reoperation (capsular release).[16–18] Nonetheless, incident data on outcome and AEs are impaired by the heterogeneity in definition and reporting.[13 19]

Valid and representative data on the safety and effectiveness of ARCR are non-existent at the Swiss national level. However, such data are paramount for optimising the indication and outcome of ARCR and for benchmarking orthopaedic clinics. Reporting standards are a prerequisite for outcome and safety data. Recently, a core outcome set (COS)[20] was defined for shoulder disorders, which includes inner core domains of pain, physical function and activities, global perceived effect (a person's assessment of their recovery or degree of improvement), and AEs.[21 22] A core event set (CES) was developed by international consensus in ARCR[12 23] and lay the ground for the current project.

Appropriate indication of ARCR and judgement on risks of AEs or unsatisfactory patient outcomes rely on validated clinical prediction tools,[24 25] which are still sparse in the field of surgical repair of a rotator cuff tear. Currently existing models focus on early surgical repair,[26] tendon healing[27 28] or shoulder functional outcomes.[29] A model for shoulder stiffness included patients with various shoulder pathologies and surgeries.[30] Furthermore, individual outcome predictions in ARCR require the identification of relevant patient and management factors. Several systematic reviews have highlighted the general lack of qualitative studies focused on prognostic factors for ARCR outcomes.[31–35] In addition, we have observed substantial heterogeneity in terms of applied methodology, core outcomes and studied prognostic factors, where certain factors (eg, age, tear size, muscle degeneration, smoking) are given greater focus over others (eg, sex, traumatic onset). The reviews highlight the need for more robust prospective studies to include additional patient-reported outcomes in a multivariable context.

## Objectives

The *overall objective* is to establish a prospective cohort of patients undergoing ARCR with standardised data collection and follow-up for the evaluation and prediction of targeted core safety, and clinical and patient-reported outcome parameters that are to be routinely collected in standard clinical care.

The *primary objective* is to develop predictive models for two core outcome parameters: (1) patient-reported Oxford Shoulder Score (OSS) functional outcome; and (2) occurrence of shoulder stiffness (primary safety event) as reported by patients and clinicians.

The *secondary objectives* are (1) to evaluate the content and applicability of the defined consensus CES (ie, ARCR CES 1.0)[23] in routine practice considering the patient's perspective; (2) to quantify the incidence of AE up to 24 months after surgery (eg, persisting or worsening pain, recurrent rotator cuff defect); (3) to validate an adapted severity classification for postoperative local AEs[12 36]; and (4) to develop predictive models for other clinically relevant outcome parameters, including patient-reported outcomes (eg, perception of improvement, return to work, return to sports, quality of life, satisfaction with surgery, acceptability of symptom state), clinical outcomes (eg, shoulder strength and motion) and specific AEs (eg, rotator cuff defect at 12 months).

## METHODS AND ANALYSIS
### Study design and setting

This is a prospective multicentre cohort study of patients undergoing ARCR with 17 participating orthopaedic centres in Switzerland and 1 German centre.

Several subprojects, associated with the main ARCR cohort study, are planned and include a systematic review of prognostic studies in ARCR, the application of the ARCR CES 1.0 for AE documentation, and the application and validation of an AE severity classification.

### Eligibility criteria

Adult patients diagnosed with a partial or full-thickness rotator cuff tear by MRI, planned for a primary arthroscopic surgical repair and giving their informed consent to participate in the cohort study will be included. Patients undergoing a specific surgical procedure for irreparable tears (ie, tendon transfer, subacromial spacer or superior capsular reconstruction), revision operations, and open or mini-open reconstructions will be excluded. Patients unable to provide written informed consent or attend clinics for follow-up visits, not fluent in German, French, Italian or English, or pregnant women will be excluded.

Patients undergoing bilateral ARCR will only be included for their first intervention.

### Intervention

Shoulder arthroscopies will be performed according to standardised clinic-specific and international guidelines[37] in the context of routine care, with patients in a beachchair or lateral decubitus position under general or local anaesthesia. The variability in the repair techniques used between clinics and surgeons will be documented. Typically, after the diagnostic arthroscopy to assess the type of rotator cuff tear (partial or full-thickness tear and involved tendons, tendon tear delamination, sign of tendon

degeneration) and concomitant injuries or lesions, the ruptured tendons are mobilised until they can be repositioned on the original footprint with as little tension as possible. Tendon fixation may be performed using one of multiple anchor and suture configurations according to the surgeon's decision. An intervention at the biceps tendon is performed if any tendinopathy or lesions to the superior labrum or biceps pulley system are observed. An anterolateral or lateral acromioplasty is performed at the surgeon's discretion, generally in the presence of a hook-shaped acromion or a critical shoulder angle larger than 35°, respectively. Operative details, including additional concomitant procedures (acromioplasty, acromioclavicular joint resection, capsulotomy, and biceps tenotomy or tenodesis) and duration of operation, are recorded immediately after surgery. A standard three-phase postoperative rehabilitation scheme is usually prescribed and will be documented in detail, including immobilisation and passive mobilisation in the first phase, active mobilisation and coordination training in the second phase, followed by the third phase of specific progressive resistance exercises.

## Outcomes

The *first primary outcome* is the patient-reported change in shoulder functional outcome between baseline and 6 months postoperatively as measured with the OSS.[38] The OSS is a condition-specific questionnaire developed for patients with a degenerative or inflammatory state of the shoulder. It contains 12 items to be answered by the patient independently, which deal with pain (degree, time point) and possible handicaps in private and professional life. There are five categories of response to every question, corresponding to a score ranging from 0 to 4. Scores are summed to give a single score ranging from 0 (worst outcome) to 48 (best outcome). Transcultural validations of this questionnaire for the German and Italian populations have been performed[39 40] and are validated for patient-based outcomes after rotator cuff repair.[41–43] While functional outcome at the last 24-month follow-up is clinically relevant, the early 6-month primary time point is chosen because of the importance in early surgical recovery and rehabilitation, particularly when considering the socioeconomic impact on professionally active patients.[4]

The *second primary outcome* is the occurrence of shoulder stiffness within 6 months after surgery: this event is poorly defined in the literature.[19] We formed a consensus definition of shoulder stiffness among specialised shoulder surgeons in a Delphi survey, which describes a postoperative restriction in passive shoulder motion diagnosed within 6 months after ARCR in at least two of the motion planes of flexion, abduction and external rotation in 0° abduction. Motion restriction is to be assessed separately for each plane according to specific threshold criteria (flexion: total motion equal to or below 90° or glenohumeral motion equal to or below 80°; abduction: total motion equal to or below 80° or glenohumeral motion

equal to or below 60°; external rotation in 0° abduction: glenohumeral motion equal to or below 20° or no more than 50% of the contralateral side value). In this project, we will identify cases of shoulder stiffness based on our consensus definition as well as clinical records and reports from clinicians and their patients.

The secondary outcomes will include (1) local AEs according to the ARCR CES, in particular the occurrence of recurrent defect of repaired tendon(s) at 12 months, when at least one repaired tendon is diagnosed with a recurrent defect by ultrasound examination, persistent or worsening pain, infection, and any local event (composite outcome); (2) functional parameters of the Constant Score (CS)[44] at 6 and 12 months, shoulder strength (kg) in abduction at 6 and 12 months, patient-reported shoulder pain on the Numeric Rating Scale (NRS) at 6, 12 and 24 months, and patient-reported shoulder function: OSS at 6, 12 and 24 months, subjective shoulder value[45] assessment at 6, 12 and 24 months; (3) general health and socioeconomic parameters including patient-reported quality of sleep (NRS) at 6, 12 and 24 months,[46] return to work, change of working condition within 6, 12 and 24 months, level of depression and anxiety at 6, 12 and 24 months based on Patient-Reported Outcomes Measurement Information System (PROMIS) scores,[47 48] patient-perceived shoulder improvement, acceptability of own symptom state,[49] quality of life (utilities and general health) at 6, 12 and 24 months using the European Quality of Life 5 Dimensions 5 Level questionnaire (EQ-5D-5L), and patient satisfaction with the surgical outcome at 12 and 24 months; and (4) safety outcome assessment, occurrence of all AEs reported by clinicians and patients (including non-local AEs within 6 months after surgery), final independent surgeon and patient-rated assessment of AEs according to perceived severity (rating scale from 0 (no complication) to 100 (death)[50]), and Comprehensive Complication Index[50] considering all AEs that occurred within 6 months after surgery.

Shoulder ultrasound examinations will be performed at 12 months by experienced clinicians independent of the operating surgeons. The repair integrity will be graded according to an adaptation of the Sugaya classification (where, when using MRI images, grade 4 or 5 defines the occurrence of a recurrent effect).[51 52] Other ultrasound parameters include the location of the recurrent defect (at the footprint/medial cuff failure), long biceps tendon status, signs of anchor displacement, and location and signs of suture cut-through.

## Participant timeline

Local investigators will identify patients who meet the eligibility criteria. Patient enrolment started on 1 June 2020 and is planned for a maximum period of 15 months. Patients will complete a preoperative evaluation no more than 2 months before surgery. Follow-up assessments will be performed at 6 weeks (±1 week) and at 6 (±1 month), 12 (±1 month) and 24 (±2 months) months postoperatively. At the final 24-month time point, only patient

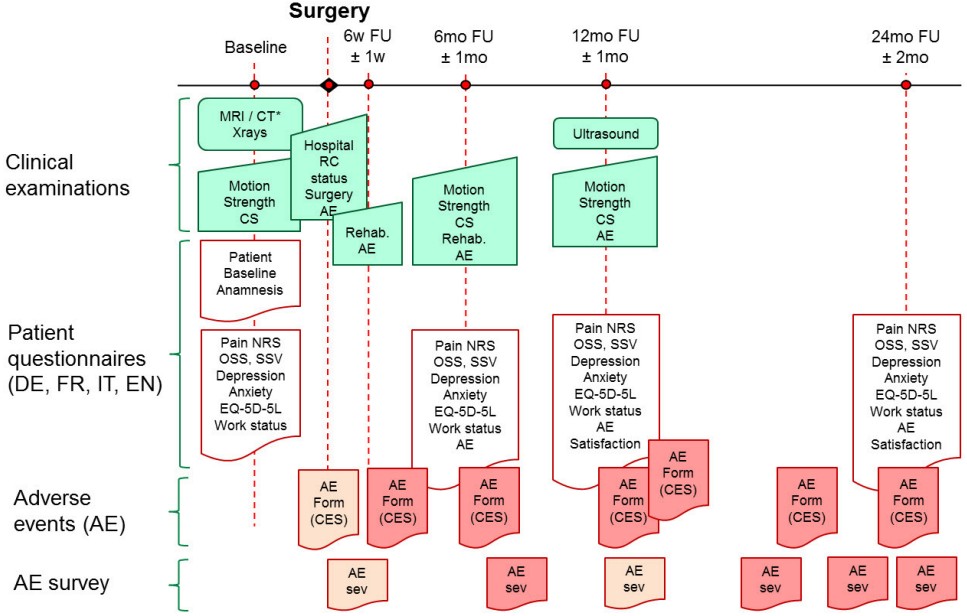

**Figure 1** Flow chart of study procedures. AE survey: surgeon and patient survey regarding AE severity (sev). Motion: shoulder range of motion. Rehab: recall on postoperative rehabilitation. *If MRI not possible. AE, adverse event; CES, core event set; CS, Constant Score; CT, arthro-CT; DE, German; EN, English; EQ-5D-5L, European Quality of Life 5 Dimensions 5 Level questionnaire; FR, French; FU, follow-up; IT, Italian; NRS, Numeric Rating Scale; OSS, Oxford Shoulder Score; RC, rotator cuff; SSV, subjective shoulder value.

self-reporting assessments, including surveys on AEs, will be documented (figure 1). The end date for the study representing the collection of the last patient questionnaire is expected on 1 November 2023.

### Baseline prognostic factors

Various baseline parameters, operative details and postoperative management variables are known or suspected to influence ARCR outcomes.[31–35]

The following patient-related factors will be recorded: patient demographics (year of birth, age, sex), socioeconomic parameters (nationality, marital status, highest level of education, employment status, last occupational position, daily physical workload), dominant side, smoking and drinking status, general physical and mental health (body mass index and obesity), American Society of Anesthesiologists classification, comorbidities (eg, diabetes), concomitant medication, level of depression and anxiety (PROMIS Depression and Anxiety Short Form 4a),[47 48] and quality of life (EQ-5D-5L).[53]

Disease-related factors are shoulder clinical examinations (pain level on NRS, range of motion, muscle strength, CS[44]), patient-reported shoulder function (see Outcomes section), radiograph parameters (critical shoulder angle,[54] acromiohumeral distance[55]), MRI or arthro-CT parameters (supraspinatus muscle atrophy,[56] tangent sign,[57] grade of fatty infiltration[58 59]) and medical history (cause of injury (trauma event), symptom duration, previous interventions (operation and timing of surgery), actual medication, extent of physical therapy).

Rotator cuff integrity and concomitant local findings will be investigated. The rotator cuff tear will be determined by MRI (or arthro-CT) and confirmed intraoperatively:

tear size, location (involved tendons) and grading (partial/complete), tendon retraction grade[60] and tear sagittal size,[61] status of the biceps tendon, and additional intraoperative observation of concomitant local injuries (superior labrum from anterior to posterior (SLAP) lesion, humeral avulsion glenohumeral ligament, Bankart lesion, humeral and glenoid-side chondral lesions).

Operative details and postoperative management will also be investigated: type of ARCR procedure (use of anchors, suture techniques), augmentation techniques (eg, platelet concentrates, scaffolds and so on),[62] additional concomitant treatment (acromioplasty, acromioclavicular joint resection, capsulotomy, biceps tenotomy or tenodesis, treatment of SLAP lesion), duration of operation, duration of hospital stay, and postoperative management (immobilisation position and duration, pain medication (eg, using non-steroidal anti-inflammatory drugs), timing of passive and active shoulder motion, and physiotherapy and muscle training).

### AE documentation and assessment process

The operating surgeons will report the occurrence of any intraoperative AE on the operation form. The occurrence of postoperative local AEs within 24 months will be reported by investigators at the clinical examination and by patients on the questionnaires. The occurrence of postoperative non-local AEs that are unrelated to the operation will be documented in a similar manner, however only within 6 months after surgery. An AE form was developed according to the ARCR CES 1.0.[23] Each AE documentation will be structured after Audigé et al[63] and includes the date/period of occurrence (intraoperative/postoperative), the affected body location (local

at the operated shoulder/non-local), the event group and specification, applied health-related intervention(s) (operative/non-operative procedure(s)), its outcome at the time of reporting (or end of the study), and the assessment of the event (causal factor(s)/severity grade/ seriousness). Severity grading will be made according to existing and adapted systems for intraoperative[64 65] and postoperative[12 36] complications.

The documentation of AEs will be checked for completeness and consistency remotely as well as on-site by reviewing selected patient files as part of the monitoring plan. A review committee (LA, AMM, TSt, HD and DS) will assess all events reported by surgeons and patients and generate queries to the respective sites as required.

Fully documented local events, including their treatment, outcome and possible causative factors, will be formulated in layman's terms and sent back to the affected patients so that they can confirm and validate collected AE data as well as assess their severity on a Visual Analogue Scale from 0 (not at all severe) to 100 (extremely severe). This subsequent rating will also be performed by their treating surgeon and four other randomly selected surgeons involved in the project, blinded to the original severity grading.

### Sample size determination

For sample size calculation, we set up a simulation study and used multiple regression to predict the change in OSS within 6 months for the most important prognostic factors. The prognostic factors were derived from an existing ARCR local registry[66] and include age, sex, body mass index/obesity status, tendon quality/degeneration and rotator cuff severity.[31–35] We accounted for type I error at 5% for statistical significance and type II error set at 20% for 80% statistical power.[67] Two thousand replications were done, and the p values were recorded to calculate the mean significance for each of the prognostic factors to reach a minimum of 80% statistical power. This approach led to a sample of 920 patients.

For the second primary outcome of shoulder stiffness, we accounted for a minimum of 10 events per variable to allow for the inclusion of a maximum of 10 predictors into the model.[68 69] The estimated event rate for shoulder stiffness from our pilot data set was 8.3%, which according to our experience might reflect an underestimation of the true rate.[17] Therefore, a 10% stiffness rate was assumed, which resulted in a sample size calculation of 900 patients.

The higher resulting number determines the final number of patients to be recruited. Therefore, 920 patients will be included with an additional 50 patients (ie, 970 patients) due to the anticipated maximum dropout rate of 5% at 6 months (based on personal experience).

### Recruitment

Study sites and local investigators were selected based on their expertise in ARCR, with support from the shoulder

and elbow expert group of Swiss Orthopaedics. Each site was visited by the project leaders to assess the adequacy of local clinical and research settings for the project as well as to ensure prior interest and commitment. The number of included sites was determined based on the reported estimate of the number of ARCR patients that could be realistically enrolled within 1 year from each site and included an allowance for overestimation (ie, all sites together estimated that they could recruit up to 40% more than the expected 970 patients within 1 year).

Patients who are enrolled after signing an informed consent form are definitively recruited for the project after documentation of baseline parameters (clinical examinations and patient questionnaires) and confirmation of ARCR during surgery. A recruitment curve is prepared every 2 weeks and sent to the project sites along with a recruitment table presenting the performance of each site. Sites that are unable to recruit the expected number of patients within the first 3 months will be considered for exclusion from the project and replaced by additional sites if the estimated total duration of patient enrolment is delayed for more than 3 months.

### Data collection methods

Data are collected on electronic or paper-based case report forms or patient questionnaires. Project parameters and used instruments are presented in the previous sections of this protocol. A training video was prepared for the collection of CS data.[44] For the measurement of shoulder muscle strength, several devices were permitted, that is, IsoForceControl (MDS Medical Device Solution, Oberburg, Switzerland), Mark-10 Force Gauge (Mark-10 Corporation, Copiague, USA), as well as hand-held (Lafayette Instrument, Lafayette, USA) or MicroFET 2 (Hoggan Scientific, Salt Lake City, USA) dynamometers; the use of a spring balance was not allowed.

Patient clinical examinations, including baseline imaging assessments, are performed at each site by experienced clinicians (who may be assisted by locally trained research staff) and documented primarily on paper-based case report forms. Baseline MRI and radiographs are coded and centralised at the University of Basel to ensure data quality control. Operative data are collected electronically by the respective surgeons shortly after surgery. Patients complete the questionnaires in their preferred language, which is limited to German, French, Italian or English, either electronically after invitation, by email or on a tablet computer at the site, or otherwise on paper. AEs are documented electronically by the respective surgeons with support from their research staff. Data collected on paper forms are entered electronically at each site or at a central location at the University of Basel based on the agreement made with each site.

### Data monitoring

A central project data manager will perform data quality control on all collected data. A flow chart will be created to describe the number of consecutively recruited patients

who had a rotator cuff repair by arthroscopic procedure or had a conversion to an open procedure, and who completed follow-up clinical and imaging examinations as well as self-reported outcome questionnaires. The reasons for patient dropout and loss to follow-up status will be monitored and described. All recorded study parameters will be described using standard descriptive statistics; continuous variables will be presented as means with SD and categorical variables as counts with percentages. The variability of data between clinics will be explored to support the identification of outlier data.

Weekly site-specific reports, including the patient enrolment list, expected follow-up timing and identification of missing, erroneous or inconsistent data, are sent to the respective local project staff. Data-related queries will be resolved remotely or by on-site monitoring visits before the final analyses are performed.

There is no plan for auditing project conduct other than via reporting at the annual meetings of the project scientific board (PSB).

### Data management

Study data will be stored using the REDCap web-based electronic data capture system[70 71] on a server that is hosted at Schulthess Klinik. REDCap conforms with Good Clinical Practice guidelines, which provide the required features for data protection and integrity, for example, password-protected access and change tracking.

Study data will be coded and exported from the REDCap system into the Stata software version 16 for statistical analyses. Data transformations and analyses will be primarily implemented using Stata and fully documented within Stata programming files. Data subsets will be prepared for analyses using alternative software (eg, R for prediction models) as appropriate.

All patients with an intraoperatively confirmed rotator cuff tear and operated by ARCR will be included in the analyses. Existing missing data will be imputed if the number of missing data is non-negligible or could potentially bias the results and conclusions.

### Systematic review of prognostic factors

A systematic review of prognostic factors for ARCR outcomes is implemented (PROSPERO registration number: CRD42020199257). Briefly, literature from 2014 to 2020 will be checked to identify longitudinal studies including patients diagnosed with a rotator cuff tear. These studies should report the effect of at least one factor on one of the following outcomes: shoulder stiffness, rotator cuff tear repair integrity and shoulder function. Data extraction will follow a predefined template and the collected data will be stored within a separate database using REDCap. Data from different studies will be described and may be synthesised depending on the data type and heterogeneity. These data will be used to generate a list of factors most likely to influence our project outcomes and therefore should be considered for inclusion in the predictive model development process.

### Predictive model development

To develop the predictive model(s), the seven steps proposed by Steyerberg *et al*[72 73] will be used. The steps comprise (1) consideration of the research question and initial data inspection; (2) coding of the predictors; (3) model specification; (4) model estimation; (5) evaluation of model performance; (6) internal validation; and (7) model presentation.

Depending on the type of outcome, different models will be fitted and evaluated, that is, multiple regression models for the change in OSS at 6 months and multivariable logistic regression models for shoulder stiffness. Model diagnostics will be performed for all models to check the underlying assumptions.

The prediction of the model(s) will be based on the baseline, operative and postoperative management variables. First, a subset of the potential prognostic factors will be defined based on whether it is thought to be most predictive. The subset will be selected separately for each outcome by the Delphi method among the investigators, whereby the factors will be noted for their known or potential prognostic value on a 5-point Likert scale from 1 (not important) to 5 (extremely important). The factors with the highest mean score among investigators will form the subset.

We will then use criterion-based procedures (eg, Akaike information criterion or adjusted $R^2$) to select the best set of predictors for the continuous outcome(s), and for the binary outcome we will use the area under the receiving operating characteristics curve.

To assess the predictive performance of the final models as well as the updated version of the prediction models, calibration plot and discrimination measures will be used. Thereby, apparent performance will be evaluated on the respective development data, and internal validated performance will be determined by bootstrapping. Independent external validation will be estimated by applying the resulting models from the development data set in the respective validation data sets. The resulting models will be used to predict the change of outcome value (ie, OSS in 6 months) and assess whether a patient will experience the event (ie, shoulder stiffness).

If we observe missing data, then missing data imputation will be performed using a method that allows for uncertainty in the imputed values (eg, multiple imputations using chained equation[74]). We will account for the clustering of records within clinics as appropriate.

### Adverse events

Occurring AEs other than those listed in the CES as well as events occurring outside the periods defined by the core set will be analysed separately for consideration of clinical relevance. This analysis will be made by the review committee and PSB comprising all local project leaders (principal investigators). Recommendations for change of the ARCR CES 1.0 by the PSB will be formulated.

The incidence of AEs, specific individual events and groups of events defined within the ARCR CES 1.0 up

to 24 months postoperatively will be displayed as the frequency of patients with an event relative to the number of patients observed, reported together with its 95% Wilson CI. These results will be presented in a summary table together with the absolute frequency. Further details on the period of occurrence will be given by stratifying for the time point of event occurrence. We will also stratify AEs according to their severity level and patient relevance. Validation of the postoperative local AE severity classification system will be implemented using previously used methods.[75][76]

## Patient and public involvement

No patient or member of the public was involved in the design of this cohort study protocol. Enrolled patients will contribute to the evaluation and validation of documented AEs and their severity grading, therefore to a potential revision of the ARCR CES. We are planning to present initial results to patients and the public and get feedback for further analyses and future model development as well as documentation system in ARCR.

## ETHICS AND DISSEMINATION
### Research ethics approval

Ethical approval was obtained on 1 April 2020 from the lead ethics committee (EKNZ, Basel, Switzerland; ID: 2019-02076).

### Protocol amendments

Minor protocol amendments, for example, database production changes to facilitate monitoring processes or improve outcome assessment by questionnaire, are fully documented. Major amendments, for example, changes to the patient information sheet and consent form, change of a local project leader or the inclusion of a new project site, will be submitted for approval by the lead ethics committee as required.

### Consent or assent

All participants will provide informed written consent prior to being enrolled into the study. The English version of the informed consent form used at the University Hospital of Basel is available as online supplemental file 1.

### Confidentiality

Project data will be handled with utmost discretion and can only be accessed by authorised personnel as outlined by a study delegation list created for each project site. Patient data will be coded, that is, identified by a unique participant number. A participant identification list will be managed and kept in a place (an electronic folder or paper-based form) only accessible to authorised staff at each site.

The project leader affirms and upholds the principle of each patient's right to privacy and that they shall comply with applicable privacy laws. In particular, anonymity of all patients shall be guaranteed when presenting the data

at scientific meetings or publishing them in scientific journals.

### Access to data

Project data will be shared at the end of the analysis process by the PSB. The Department of Clinical Research (German Departement Klinische Forschung, DKF) at the University Hospital of Basel will act as an independent data access committee and will store the data at the time of publication on secure servers, maintained and backed up by the Information and Communication Technology Department at the University Hospital of Basel. Researchers who wish to reuse data will be able to submit a project synopsis to the DKF at dkf.unibas.ch/contact. A data-sharing statement referring researchers to the DKF for data access will be disseminated in the publications. Metadata describing the type, size and content of the data sets will be shared along with the study protocol on the Harvard Dataverse repository available online (https://dataverse.harvard.edu/). Additionally, the case report forms will be uploaded on a medical data models portal (https://medical-data-models.org/) and all variables will be annotated by their Unified Medical Language System Concept Unique Identifier to improve accessibility to other clinicians.

### Dissemination policy

This project will lead to multiple open-access, peer-reviewed scientific publications, which will be prepared according to international standards (eg, the Strengthening the Reporting of Observational Studies in Epidemiology statement[77] for cohort studies; Transparent Reporting of a multivariable prediction model for Individual Prognosis or Diagnosis[78] statements for prognostic studies; Preferred Reporting Items for Systematic Reviews and Meta-Analyses[79] statement for systematic reviews). Publication authorship will be regulated according to the guidelines of the Swiss Academies of Arts and Sciences.[80] Results will be submitted for presentation at national and international conferences. In addition, lay summary results will be developed and made available for patients and the public.

### Scientific relevance and broader impact

This project initiates the development of personalised risk predictions to support the surgical decision process in ARCR. The consensus CES may become an international reference for the reporting of complications in clinical studies and registries, and may therefore provide a solid metric for the documentation of surgical safety in ARCR. Methodological insight gained from this project will be easily transferable to similar initiatives and thus may foster the realisation of other cohorts on safety and effectiveness outcome in shoulder surgery (eg, arthroplasty) and orthopaedics in general.

For patients affected by rotator cuff tears and their surgeons, this study will be the first to provide solid data on the incidence of patient-validated AEs and other core

outcomes up to 2 years after surgical repair based on international consensus COS and CES. This study will allow the investigation of a comprehensive list of potential prognostic factors to generate predictive models for these core outcomes and hence offer personalised health information to support future patients and surgeons in the decision process for surgery. Outcome predictors and risk calculators are increasingly being developed in numerous medical fields including surgery and orthopaedics, and they are in development in the field of ARCR.

This study will assess the structure and content of the ARCR CES and consolidate its validity in capturing unfavourable events of importance to both patients and surgeons; considering the patient's perspective is an essential step in the development of a COS. Furthermore, the validation of an adapted severity classification of AEs in this study will provide an essential system for assessing surgical morbidity in orthopaedics. We expect that the ARCR CES and the event severity classification will become international standards for the reporting of ARCR AEs in clinical studies and registries, and therefore provide a solid metric for the documentation of surgical safety in ARCR.

This study fosters the enterprise in developing a Swiss-wide registry of ARCR, which will allow the ongoing evaluation and prediction of targeted core safety and clinical and patient-reported outcomes. The identification of factors mostly associated with relevant outcomes will facilitate a lean and straightforward documentation process for ARCR patients in Switzerland and abroad.

**Author affiliations**
[1]Research and Development, Schulthess Klinik, Zurich, Switzerland
[2]Orthopaedic Surgery and Traumatology, University Hospital Basel, Basel, Switzerland
[3]Basel Institute for Clinical Epidemiology and Biostatistics, University Hospital Basel, Basel, Switzerland
[4]Swiss Patient Safety Foundation, Zurich, Switzerland
[5]Institute of Social and Preventive Medicine, University of Bern, Bern, Switzerland
[6]Medical Communication/Psychosomatic Medicine, University Hospital Basel, Basel, Switzerland
[7]Trauma and Ortho Unit, Ospedale Regionale di Lugano, Lugano, Switzerland
[8]Shoulder Center, Hirslanden Clinique La Colline, Geneva, Switzerland
[9]Division of Orthopaedics and Trauma Surgery, Department of Surgery, Geneva University Hospitals, Geneve, Switzerland
[10]Klinik Gut Sankt Moritz, Saint Moritz, Switzerland
[11]Clinic for Orthopaedics and Traumatology, Baden Cantonal Hospital, Baden, Switzerland
[12]Center for Orthopaedics and Neurosurgery, In-Motion, Wallisellen, Switzerland
[13]Clinic for Orthopaedic Surgery and Traumatology of the Musculoskeletal System, Cantonal Hospital of St.Gallen, St Gallen, Switzerland
[14]Division of Orthopaedics and Trauma Surgery, La Tour Hospital, Meyrin, Switzerland
[15]Service for Orthopaedics and Traumatology of the Musculoskeletal System, Hôpital du Valais - Centre Hospitalier du Valais Romand, Martigny, Switzerland
[16]Department of Shoulder and Elbow Surgery, Center for Musculoskeletal Surgery, Charité Medicine University, Berlin, Germany
[17]Shoulder and Elbow Center, Arthro Medics, Basel, Switzerland
[18]Department of Orthopaedic Surgery and Traumatology, Inselspital, Bern University Hospital, University of Bern, Bern, Switzerland
[19]Shoulder and Elbow Surgery, Schulthess Klinik, Zurich, Switzerland
[20]Center for Endoprosthetics and Joint Surgery, Endoclinic, Zürich, Switzerland
[21]Orthopaedic Shoulder and Elbow, Canton Hospital Basselland, Bruderholz, Switzerland
[22]Department of Orthopaedics, Balgrist University Hospital, University of Zurich, Zurich, Switzerland
[23]Shoulder, Elbow and Orthopaedic Sports Medicine, Orthopaedics Sonnenhof, Bern, Switzerland
[24]Campus SLB, Swiss Institute for Translational and Entrepreneurial Medicine, Stiftung Lindenhof, Bern, Switzerland

**Acknowledgements** The authors acknowledge the support of Dr Martina Gosteli, medical librarian at the University of Zurich, for implementing the preliminary literature database search to support this protocol development. The authors would also like to acknowledge the support of Dr Melissa Wilhelmi, medical writer at Schulthess Klinik, Zurich, Switzerland, for manuscript proof-reading.

**Collaborators** ARCR_Pred Study Group: Jannine Buchschacher, Lena Fankhauser, Corinne Eicher, Gernot Willscheid (ARTHRO Medics, Basel, Switzerland (ART)); Doruk Akgün, Kathi Thiele, Marvin Minkus, Victor Danzinger, Katrin Karpinski (Charité Medicine University, Berlin, Germany (BER)); Sebastian Mueller, Claudia Haag-Schumacher (Canton Hospital Basselland, Bruderholz, Switzerland (BRU)); Viviane Steffen, Sarah Fournier, Deborah Marietan, Sebastien Pawlak (Centre Hospitalier du Valais Romand, Martigny, Switzerland (CHV)); Britta Hansen, Ferdinand Lovrek, Marco Zanetti, Nadja Mamisch (Endoclinic, Zürich, Switzerland (END)); Christian Steiner, Georg Ahlbäumer, Jakob Bräm, Jens Fischer, Alexander Delvendahl (Klinik Gut, St Moritz, Switzerland (GUT)); Patricia Simao, Abed Khourani (Hirslanden Clinique la Colline, Geneva, Switzerland (HIR)); Anne-Sophie Foucault, Frank Kolo (La Tour Hospital, Meyrin, Switzerland (HUG)); Adrian Schenk, Johannes Weihs, Remy Flückiger, Philipp Scacchi, Paolo Lombardo (Inselspital, Bern University Hospital, Bern, Switzerland (INB)); Larissa Hübscher, Ralph Berther, Christine Ehrmann (In-Motion, Wallisellen, Switzerland (INM)); Raffaela Nobs, Richard Niehaus, Nisha Grünberger, Philipp Kriechling, Susanne Bensler (Cantonal Hospital of Baden, Baden, Switzerland (KSB)); Hans-Kaspar Schwyzer, Fabrizio Moro, Michael Glanzmann, Barbara Wirth, Christian Jung, Florian Freislederer, Manuela Nötzli, Frederik Bellmann, Franz Anne, Jörg Oswald, Alex Marzel, Cécile Grobet, Marije de Jong, Martina Wehrli, Jan Schätz (Schulthess Klinik, Zurich, Switzerland (KWS)); Francesco Marbach, Marco Delcogliano, Davide Previtali, Florian Schönweger, Elena Porro, Gabriela Induni-Lang, Giuseppe Filardo, Filippo Del Grande, Pietro Feltri, Schiavon Guglielmo (Ospedale Regionale di Lugano, Lugano, Switzerland (LUG)); Christian Spross, Martin Olach, Michael Badulescu, Vilijam Zdravkovic, Stephanie Lüscher, Jörg Scheler, Lena Öhrström (Cantonal Hospital of St.Gallen, St Gallen, Switzerland (SGA)); Annabel Hayoz, Frederick Schuster, Julia Müller-Lebschi (Orthopädie Sonnenhof, Berne, Switzerland (SON)); Christian Gerber, Samy Bouaicha, Paul Borbas, Anita Hasler, Florian Grubhofer, Sabrina Catanzaro, Sabine Wyss, Reto Sutter (Balgrist University Hospital, Zurich, Switzerland (UKB)); Mohy Taha, Cornelia Baum, Ilona Ahlborn, Simone Hatz, Giorgio Tamborrini-Schütz (University Hospital of Basel, Basel, Switzerland (USB)); Christian Appenzeller-Herzog (University Library Basel, Basel, Switzerland).

**Contributors** LA and AMM are the initiators and project leaders. LA, AMM, HCCB, SA, DS and SH were involved in the study design, which was reviewed and commented by principal investigators CC, GC, HD, KE, MF, BJ, AL, BKM, PM, CR, MScha, MSche, CS, TSu, KW and MZ. Preparation of the manuscript was done by LA, AMM, SA and TSt. HCCB, DS and SH edited and critically revised the paper. All authors have read and approved the manuscript. LA is the guarantor of the manuscript. This is an investigator-initiated project at the University Hospital of Basel. The principal investigator and project leader (AMM) is the official sponsor representative for the project and was involved in all phases of the project, from its conception to the current implementation steps. The project initiators and project leaders (LA and AMM) have ultimate authority over any of the project activities. A project scientific board (PSB) comprises the project leaders (LA and AMM), project investigators at each site (CC, GC, HD, KE, MF, BJ, AL, BKM, PM, CR, MScha, MSche, CS, TSu, KW and MZ) and project partners (SA, HCCB, DS and SH). The PSB shall meet at specific time points during the study—before the study start, after completion of recruitment and the 12-month follow-up—and at the end of the study. The agenda of these meetings will focus on (however is not limited to) patient enrolment and the completion of follow-up examinations and questionnaires, the documentation process in REDCap, data quality issues (completeness and consistency), monitoring activities, adverse event assessment and management, baseline patient description, ranking of prognostic factors for prognostic models, progress of data analysis, publication strategy and decisions regarding data sharing. Between these meetings, communication will be maintained between the project coordinating team and investigators via various channels, including emails, quarterly newsletters, phone calls and (video) conference calls as required.

**Funding** This project is funded by the Swiss National Science Foundation (SNF Project ID 320030_184959, http://p3.snf.ch/project-184959). A complementary grant was provided by Swiss Orthopaedics. The sites Charitè Medicine University, Berlin, Germany (BER) and University Clinic Balgrist, Zurich, Switzerland (UKB) are funding their own participation in the project.

**Competing interests** None declared.

**Patient consent for publication** Obtained.

**Provenance and peer review** Not commissioned; externally peer reviewed.

**ORCID iDs**
Laurent Audigé http://orcid.org/0000-0003-3962-3996
David Schwappach http://orcid.org/0000-0001-8668-3065

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
