## [Reviewer comments · BMJ Open]

ARTICLE DETAILS

TITLE (PROVISIONAL)	Protocol for the ARCR_Pred cohort-study: Swiss-wide multicenter evaluation and prediction of core outcomes in arthroscopic rotator cuff repair
AUTHORS	Audigé, Laurent; Bucher, Heiner; Aghlmandi, Soheila; Stojanov, Thomas; Schwappach, David; Hunziker, Sabina; Candrian, Christian; Cunningham, Gregory; Durchholz, Holger; Eid, Karim; Flury, Matthias; Jost, B; Ladermann, Alexandre; Moor, Beat; Moroder, Philipp; Rosso, Claudio; Schär, Michael; Scheibel, Markus; Spormann, Christophe; Suter, Thomas; Wieser, Karl; Zumstein, Matthias; ARCR_Pred Study Group, Name; Müller, Andreas

VERSION 1 – REVIEW

REVIEWER	Babhulkar, Ashish Deenanath Mangeshkar Hospital and Research Centre, Shoulder & Sports injuries
REVIEW RETURNED	20-Jan-2021

GENERAL COMMENTS	Dr. Andreas Müller has designed an excellent study which is watertight for any bias. Study will provide useful evidence on outcomes of rotator cuff repair. There are 3 minor corrections, mainly on usage of appropriate terminology and minor disclosures to patients. Both the disclosures that I have suggested do not affect the outcome or technique of research. The purpose of disclosures was to better counsel the patient on the exact nature of intervention. These are not binding to final approval.
--

REVIEWER	Pripp, Are Hugo Oslo universitetssykehus Ullevål, Oslo Centre for Biostatistics & Epidemiology
REVIEW RETURNED	11-Mar-2021

GENERAL COMMENTS	The protocol paper is clearly written and described. I recommend it for publications. My only comment is that they consider to publish all computer codes (i.e. stata or R syntaxes etc.) for the prognostic models.
--

VERSION 1 – AUTHOR RESPONSE

Reviewer: Reviewer: 1
Reviewer Name: Dr. Ashish Babhulkar
Institution and Country: Deenanath Mangeshkar Hospital and Research Centre

Comments	Author Responses	Changes made
Dr. Andreas Müller has designed an excellent study which is watertight for any bias. Study will provide useful evidence on outcomes of rotator cuff repair.	We thank the reviewer very much for appreciating the quality of our study.	
There are 3 minor corrections, mainly on usage of appropriate terminology and minor disclosures to patients. Both the disclosures that I have suggested do not affect the outcome or technique of research. The purpose of disclosures was to better counsel the patient on the exact nature of intervention. These are not binding to final approval.	We agree about the importance of terminology and therefore the suggestion of the reviewer is appropriate. Indeed the term “tendinosis” is not synonym to “rotator cuff tear”. The English version of the patient information and informed consent form (PIC) is an official translation from the officially-approved German version in which the term “Sehnenrissen (= Rotatorenmanschettenrissen)” was used, yet we overlooked this mistake. We agree that this translation is not adequate, and therefore corrected this form to a new Version 2 (dated 17.03.2021), which we will submit also to our local EC for potential use at the few involved sites using a PIC in English.	We replaced the word “tendinosis” by “tear of the shoulder muscle tendons” in 4 locations within the provided Informed Consent Form, hence making an amended version 2 (dated 17.03.2021).

Reviewer: 2

Reviewer Name: Dr. Are Hugo Pripp

Institution and Country: Oslo universitetssykehus Ullevål

Comments	Author Responses	Changes made
The protocol paper is clearly written and described. I recommend it for publications.	Thank you for finding our manuscript well-written.	
My only comment is that they consider to publish all computer codes (i.e. stata or R syntaxes etc.) for the prognostic models.	Research activities are becoming increasingly transparent, including full disclosure and sharing of study protocols, electronic databases, data and occasionally statistical codes. The methodology we are using for generating the prognostic models is published (Steyerberg et al.) and many codes are already available for the purpose of prognostic model development (e.g. using R). As part of our agreement with the Swiss National Science Foundation, we will share our database after a period of publication by the project team. Of course we will publish the final regression models in details along with the respective publications. We do not plan at this stage to openly share the statistical programming codes themselves, outside specific requirements from targeted scientific journals. This is a suggestion we can understand very well and this decision would require to be made by our scientific board in due time. We may well consider code sharing at least in the context of a research cooperation.	No change made